# Surfing motility is a complex adaptation dependent on the stringent stress response in *Pseudomonas aeruginosa* LESB58

**Daniel Pletzer**[1,2☯]*, **Evelyn Sun**[1☯], **Caleb Ritchie**[1], **Lauren Wilkinson**[1], **Leo T. Liu**[1], **Michael J. Trimble**[1], **Heidi Wolfmeier**[1], **Travis M. Blimkie**[1], **Robert E. W. Hancock**[1]*

**1** Centre for Microbial Diseases and Immunity Research, Department of Microbiology and Immunology, University of British Columbia, Vancouver, Canada, **2** Department of Microbiology and Immunology, University of Otago, Dunedin, New Zealand

☯ These authors contributed equally to this work.
* daniel.pletzer@otago.ac.nz (DP); bob@hancocklab.com (REWH)

**Data Availability Statement:** All fastq and count files are available under Gene Expression Omnibus (GEO) accession number GSE138716. The cDNA library for each sample was sequenced twice to

## Abstract

Cystic fibrosis (CF) is a genetic disease that affects mucin-producing body organs such as the lungs. Characteristic of CF is the production of thick, viscous mucus, containing the glycoprotein mucin, that can lead to progressive airway obstruction. Recently, we demonstrated that the presence of mucin induced a rapid surface adaptation in motile bacteria termed surfing motility, which data presented here indicates is very different from swarming motility. *Pseudomonas aeruginosa*, the main colonizing pathogen in CF, employs several stress coping mechanisms to survive the highly viscous environment of the CF lung. We used motility-based assays and RNA-Seq to study the stringent stress response in the hypervirulent CF isolate LESB58 (Liverpool Epidemic Strain). Motility experiments revealed that an LESB58 stringent response mutant (Δ*relA*Δ*spoT*) was unable to surf. Transcriptional profiling of Δ*relA*Δ*spoT* mutant cells from surfing agar plates, when compared to wild-type cells from the surfing edge, revealed 2,584 dysregulated genes. Gene Ontology and KEGG enrichment analysis revealed effects of the stringent response on amino acid, nucleic acid and fatty acid metabolism, TCA cycle and glycolysis, type VI secretion, as well as chemotaxis, cell communication, iron transport, nitrogen metabolic processes and cyclic-di-GMP signalling. Screening of the ordered PA14 transposon library revealed 224 mutants unable to surf and very limited overlap with genes required for swarming. Mutants affecting surfing included two downstream effector genes of the stringent stress response, the copper regulator *cueR* and the quinolone synthase *pqsH*. Both the *cueR* and *pqsH* cloned genes complemented the surfing deficiency of Δ*relA*Δ*spoT*. Our study revealed insights into stringent stress dependency in LESB58 and showed that surfing motility is stringently-controlled via the expression of *cueR* and *pqsH*. Downstream factors of the stringent stress response are important to investigate in order to fully understand its ability to colonize and persist in the CF lung.

obtain greater sequencing depth. The same steps for QC, alignment, and counts were followed in both cases. Counts were merged for each sample prior to all analyses. Library sizes representing merged counts from the two sequencing runs had a minimum of 670,007, median of 1,209,676, and maximum of 2,044,247. The full list of differentially expressed genes is included in S1 Data.

**Funding:** Research reported in this publication was supported by grants from the Cystic Fibrosis Canada award number 2585 and Canadian Institutes for Health Research grant FDN-154287. DP is supported by an Alexander von Humboldt Feodor Lynen Postdoctoral Fellowship, a Cystic Fibrosis Canada Postdoctoral fellowship, and a fellowship from the Michael Smith Foundation for Health Research. CTR received a Summer Studentship award from the Centre for Blood Research, HW received an Early Postdoc Mobility fellowship from the Swiss National Science Foundation, and REWH holds a Canada Research Chair in Health and Genomics and a UBC Killam Professorship. The funders had no role in study design, data collection and analysis, decision to publish, or preparation of the manuscript.

**Competing interests:** The authors have declared that no competing interests exist.

## Author summary

Cystic fibrosis (CF) is a progressive disease associated with excessive mucus build up in the lungs, blocking airways and facilitating bacterial persistence. CF affects >30,000 people in the US and currently there exists no cure. Bacterial infections such as the ones that involve *Pseudomonas aeruginosa* worsen treatment strategies because of the pathogens' ability to evade not only the immune system but also antibiotic killing. In the CF lung, *P. aeruginosa* has been proposed to employ rapid surface motility adaptations in the form of swarming and the more recently identified surfing motility. Here, we demonstrate that surfing is indeed a novel form of motility that can be distinguished from swarming, and further demonstrate that surfing is dependent on the stringent stress response in an aggressive *P. aeruginosa* CF isolate. We discovered two novel downstream factors of the stringent response, a copper regulator and a quinolone synthase, that contribute to mediating surfing motility. Investigating adaptive behaviors such as surface motility is important to understand how these contribute to host-pathogen interactions.

## Introduction

*Pseudomonas aeruginosa* is a ubiquitous Gram-negative pathogen that causes opportunistic and difficult-to-treat infections. It is strongly associated with chronic debilitating lung infections in individuals with cystic fibrosis (CF) [1]. A hallmark of pathogenic *P. aeruginosa* is their evolutionary adaption to challenging host conditions such as those found in the CF lung [2]. A particularly aggressive CF isolate is the transmissible Liverpool Epidemic Strain (LES), originally isolated from the sputum of a CF patient [3]. LES isolates are associated with enhanced virulence and greater patient morbidity [4], predominance in intra-species competition [5], stronger persistence in the bronchial lumen [6], increased biofilm formation, decreased motility, and elevated levels of antibiotic resistance [2].

*P. aeruginosa* possesses several virulence factors that facilitate invasion of the host, compromise immune defences, and ensure its survival and colonization in the host environment. To establish an infection, it has been proposed that *Pseudomonas* type IV pili are involved in binding to the apical epithelial surface, while flagellar-mediated motility is proposed to be important for the colonization of the basolateral surface of airway epithelium [7]. The type of motility can vary substantially depending on the viscosity and composition of the environment [8]. Recently, Abdullah *et al.* [9] showed that the reduced water content of the CF airways can trigger the misfolding of the MUC5B mucin protein that further contributes to the thick, viscous gel-like properties of the mucus in the CF lung. At the viscosity encountered in the mucous layer of the CF lung, *Pseudomonas* likely adopts complex surface growth and motility behaviours, such as swarming and biofilm formation using flagella and type-IV pili [10] or flagella-mediated surfing in the presence of the glycoprotein mucin [11].

*P. aeruginosa* employs a variety of adaptation mechanisms to cope with environmental stresses [12]. The stringent stress response signaling pathway, mediated by the second messenger molecule guanosine (penta)tetraphosphate (p)ppGpp, is an evolutionarily conserved response that is activated in various cellular stress conditions that restrict bacterial growth (e.g. nutrient or iron limitation). In *P. aeruginosa*, (p)ppGpp is synthesized through the two enzymes RelA and SpoT. Both enzymes can transfer pyrophosphate from ATP onto GTP to form guanosine pentaphosphate, which is rapidly converted to ppGpp. The accumulation of ppGpp remodels the RNA polymerase that leads to a switch from energetically-costly processes to energy-saving and stress coping mechanisms [13]. The stringent stress response has

been previously studied in various *Pseudomonas* human and plant isolates and shown to be associated with twitching and swarming motility, biofilm formation, hemolytic capacity and abscess formation as well as growth in a murine skin infection model [6, 14–18].

The CF lung is considered a stressful environment due to increased oxidative stress levels [19], chronic activation of the innate immune system [20], and bacterial competition upon antibiotic exposure [12]. Therefore, here we aimed to characterize the role of the stringent stress response in relation to motility of the hypervirulent CF epidemic isolate LESB58. We present evidence that the stringent stress response is required for *P. aeruginosa* to adapt to environmentally challenging conditions. This stress response was shown to link the complex adaptive motile behaviour, surfing, with the strong involvement of the stringently-controlled copper regulator *cueR* and quinolone synthase *pqsH*.

## Results

### The *P. aeruginosa* LESB58 stringent response mutant was impaired in surfing

We confirmed literature reports [14, 15, 21] showing that the stringent response was required for swarming, adherence and biofilm formation, as well as pyoverdine and pyocyanin production, in the clinical isolate *P. aeruginosa* LESB58 (S1 Text, S1 Fig). Vogt *et al.* [14] demonstrated that ppGpp was not required for swimming in *P. aeruginosa* PAO1, as indeed we confirmed (S2 Fig), and despite the rather poor swimming motility of strain LESB58, ppGpp did not appear to be required for the weak swimming motility observed for this strain either (S1 Text, S2 Fig). To determine the role of the stringent stress response in other adaptive behaviours, we further investigated surfing motility. Surfing is a conserved form of surface motility that occurs when mucin, a high molecular weight glycoprotein present in mucus, is added to the medium [22]. It exhibits clear differences from swarming surface motility in that it is dependent on flagella but not pili or rhamnolipids (required for *P. aeruginosa* swarming), and can occur at a wide range of substrate viscosities under both nutrient-limiting or rich conditions (restricted in the case of swarming), and with certain other lubricating agents in place of mucin [11, 22, 23]. At the molecular level, surfing cells exhibit differential expression of more than a thousand genes, including many regulators, when compared to swimming, swarming and sessile cells [23].

Intriguingly, LESB58 showed excellent surfing capability (Fig 1). In contrast, a double deletion mutant of *relA* and *spoT* in the LESB58 background, Δ*relA*Δ*spoT*, was defective in surfing.

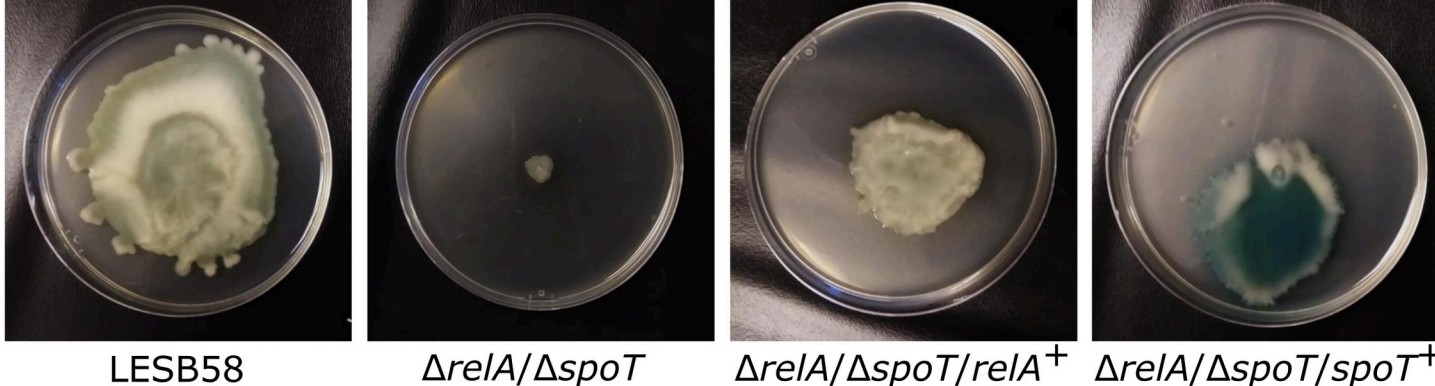

LESB58 Δ*relA*/Δ*spoT* Δ*relA*/Δ*spoT*/*relA*⁺ Δ*relA*/Δ*spoT*/*spoT*⁺

**Fig 1. Surfing motility in *P. aeruginosa* LESB58 and Δ*relA*/Δ*spoT*.** Surfing on KB 0.3% agar plates with 0.4% mucin of the LESB58 wild-type, Δ*relA*Δ*spoT* stringent response mutant, and *relA* and *spot* complemented strains. The strains were grown for 24 h at 37°C. Experiments were repeated 3–5 times with similar results.

This could be partially complemented with either the *spoT* or *relA* cloned genes, although the *spoT* complement was much more highly pigmented (Fig 1). There was no basic defect in flagella production due to any of these mutations, since we observed similar swimming zone sizes after 24 h for WT and all mutants in both the strains LESB58 and PAO1 (although the swimming colony morphology for strain LESB58 was somewhat unusual). Furthermore, all mutants appeared to show similar swimming ability when viewed in aqueous suspension using a light microscope. Nevertheless, to investigate if the loss of surfing involved flagella loss, we examined cell morphology of mutant cells on surfing agar plates and compared them to wild-type cells using transmission electron microscopy (TEM). The LESB58 wild-type showed two distinct morphologies with electron-dense thin rods and less dense enlarged cells at the edge and center, while the stringent response mutant only exhibited enlarged cells that were both longer and wider than wild-type cells (S3 Fig). Flagella could only be observed on cells at the centre of surfing plates, but not at the edge for WT, as previously described [11]. We were unable to detect flagella in the spots formed by the Δ*relA*Δ*spoT* double mutant (S4 Fig), although the large amount of extracellular material under these circumstances made observations difficult.

Since surface motility adaptations (i.e., swim, swarm, surf) utilize flagella for colony propagation, we examined the expression of some genes involved in flagella synthesis, chemotaxis, and quorum-sensing using qRT-PCR (Table 1). RNA was isolated from *P. aeruginosa* LESB58 stringent response mutant cells and compared to wild-type cells from the motility edge. Under swarming conditions, downregulation in the Δ*relA*Δ*spoT* stringent response mutant was observed for the chemotaxis gene *cheY* by 3.8-fold, the two-component response regulator of flagella synthesis *fleR* by 3.5-fold, and the rhamnosyltransferase *rhlB* (which is involved in the synthesis of rhamnolipid, a surfactant required for swarming) by 5.8-fold. We also observed downregulation of the quorum-sensing regulators *lasR* and *rhlR* by 4.3- and 3.5-fold respectively. In contrast under surfing conditions, most of these genes were relatively upregulated or unaltered in expression in the double mutant (Table 1).

## The LESB58 stringent stress response regulated surfing motility

To further understand the surfing deficiency of the stringent response mutant, the transcriptional profile of the LESB58 Δ*relA*Δ*spoT* (non-surfing) mutant cells from surfing agar plates was compared to wild-type cells isolated from the edge of the surfing colony. This revealed the

**Table 1. Alterations in LESB58 Δ*relA*/Δ*spoT* mRNA transcripts under surfing, swarming, and swimming conditions.** The stringent response double mutant was compared to the edges of wild-type motility zones. All experiments were performed on KB agar plates.

| Gene | Product Description | Fold change in Δ*relA*/Δ*spoT* compared to WT | | |
|---|---|---|---|---|
| | | Surfing | Swarming | Swimming |
| *fleQ* | Transcriptional regulator FleQ, a c-di-GMP responsive regulator that influences flagella synthesis | 1.7 | -1.0 | 1.1 |
| *fleR* | Transcriptional response regulator FleR that controls flagella synthesis | 1.1 | -3.5 | -1.4 |
| *cheY* | Two-component response regulator CheY that modulates chemotaxis | 1.7 | -3.8 | -1.5 |
| *rhlB* | Rhamnosyltransferase chain B involved in the production of rhamnolipid surfactant required for swarming | 3.2 | -5.8 | -2.1 |
| *lasR* | Transcriptional regulator LasR for the quorum sensing system based on N-3-oxo-dodecanoyl-L-homoserine lactone | 2.1 | -4.3 | 1.8 |
| *rhlR* | Transcriptional regulator RhlR for the quorum sensing system based on N-butanoyl-L-homoserine lactone | 2.2 | -3.5 | -2.2 |
| *pqsH* | FAD-dependent monooxygenase involved in synthesis of the quorum sensing effector *Pseudomonas* Quinolone Signal (PQS) | -2.8 | -2.4 | -1.1 |
| *pqsR / mvfR* | Transcriptional regulator MvfR modulates quorum sensing | 1.7 | 1.3 | -1.0 |
| *cueR* | Cu(I)-responsive transcriptional regulator CueR | -4.4 | -1.9 | -3.1 |

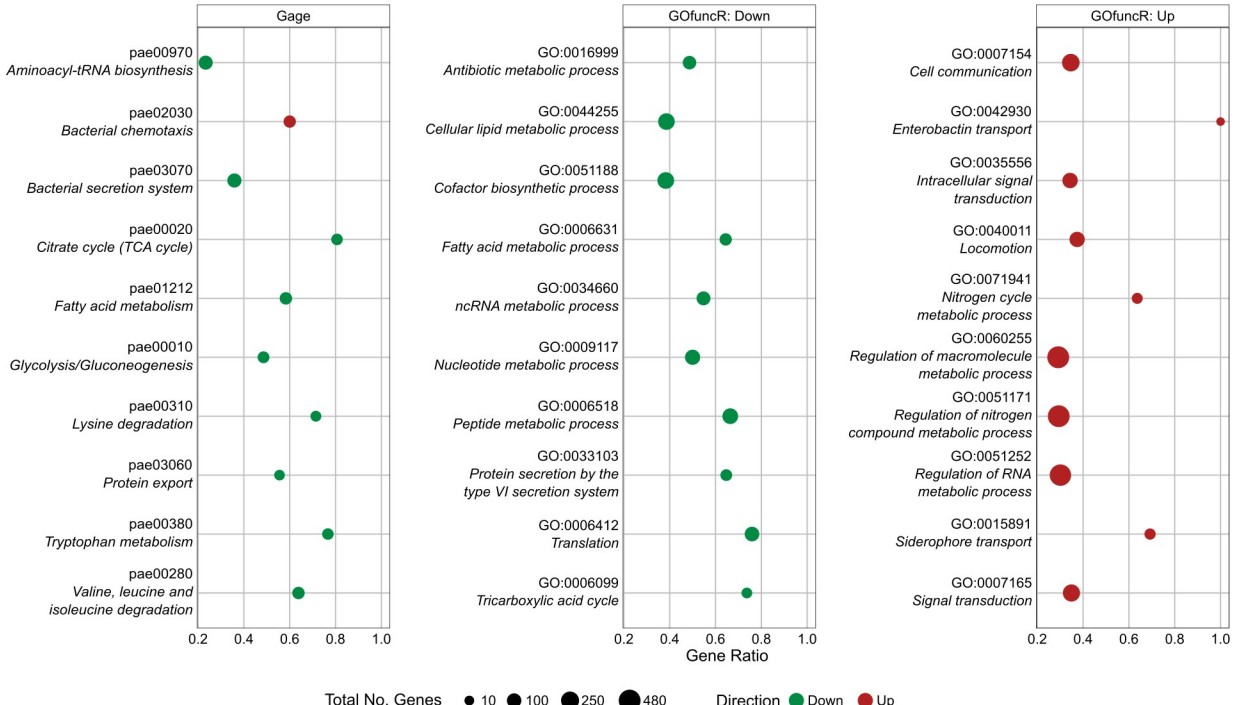

**Fig 2. KEGG analysis and GO enrichment of differentially expressed genes comparing the *P. aeruginosa* LESB58 stringent stress response mutant Δ*relA*/Δ*spoT* to the wild-type strain under surfing conditions.** The gene ratio on the x-axis represents the proportion of genes in the particular pathway/functions category that were dysregulated. A) KEGG enrichment of differential gene expression performed by GAGE analysis with a threshold of q-value ≤ 0.1. B, C) Results of selected GO term enrichment for Biological processes performed by GOfuncR on the list of differentially expressed genes with downregulated (B) and upregulated (C) GO terms. GO terms were considered significant with q-value ≤ 0.1. A-C) Dot size indicates total number of genes annotated to a particular term / pathway.

massive dysregulation of 2,584 genes (±1.5-fold change, adjusted *p*-value<0.05), 43.6% of the genome, comprising 1,261 upregulated and 1,323 downregulated genes. KEGG pathway (Fig 2A) analysis and Gene Ontology (GO) enrichment (Fig 2B and 2C), revealed that downregulated Δ*relA*Δ*spoT* mutant genes were involved in amino acid, nucleic acid and fatty acid metabolism, TCA cycle and glycolysis, as well as type VI secretion. On the other hand, chemotaxis, cell communication, iron transport, and nitrogen metabolism were upregulated. A closer look at individual genes required for quorum sensing (QS) revealed that most genes in this pathway were downregulated (S5 Fig). Intriguingly, a significant upregulation of many genes involved in flagella biosynthesis and chemotaxis was evident (S6 Fig), which was not consistent with the explanation that a loss of flagella gene expression was responsible for the phenotype of the stringent response double mutant. Consistent with decreased adherence (S1C and S3B Figs), downregulation of many pilus genes, including a 11.4-fold decrease in the structural *pilA* gene, was observed (S7 Fig).

Furthermore, the expression of genes encoding 21 of the 38 cyclic-di-GMP modulating enzymes identified in *P. aeruginosa* were regulated by ppGpp as determined by their up (18 genes) or down (3 genes) regulation in the Δ*relA*Δ*spoT* stringent response mutant. These genes included PA0285, PA0290, PA0575, PA0847, *rbdA*, *roeA*, PA1181, PA1851, PA2072, PA2771, PA2870, PA3258, *nbdA*, PA3825, *rocR*, *bifA*, PA4396, *gcbA*, PA4959, *dipA*, and PA5442. Other related genes included the HD-Gyp PA2572, the PilZ regulators PA0012, PA2799, PA2989 and PA4608, and the MshEN PA3740 (all upregulated) (S8 Fig).

## Identification of genes required for surfing motility and stringent dependency

To further characterize the genes that might underlie the effects of the stringent response double mutant, a group of genes that were required for surfing motility was determined by screening the ordered PA14 transposon library [24] under surfing conditions. This screen revealed 224 transposon mutants, including 25 regulatory mutants, that were either defective in surfing or had a non-surfing phenotype (S1 Table). Intriguingly there was very little overlap with mutants identified from an analogous screen performed to identify mutants involved in swarming [25].

To identify downstream mediators of the stringent stress response involved in surfing in LESB58, we focused on regulatory genes in the ppGpp-deficient Δ*relA*Δ*spoT* mutant. To extend this analysis, qRT-PCR was utilized to examine gene expression of 11 of the regulators for which mutants had a surfing defect (Table 2). Dysregulation of 9 of these by more than 2-fold was demonstrated and of these, four were demonstrated to be downregulated, including the copper-transport regulator *cueR*, a positive regulator of copper metabolism [26], that was strongly down regulated by 4.4-fold (Table 2).

Interestingly, we observed upregulation in the stringent response double mutant of the two quorum sensing regulators *rhlR* and *pqsR*, which when mutated lead to surfing deficiency [22, 23] (S1 Table) and also mediate stress responses [12, 27]. Since both are involved in the regulation of *Pseudomonas* quinolone signal (PQS) synthesis during stress [27], downstream PQS genes were further investigated and it was shown that the *pqsABCDE* operon, *pqsH* and *phnAB* were all downregulated (by -1.6 to -3.5-fold) in the mutant (S5 Fig). The gene expressing FAD-dependent monooxygenase *pqsH*, involved in the last step of PQS synthesis, was confirmed by qRT-PCR to be 2.8-fold downregulated in the Δ*relA*Δ*spoT* mutant.

To determine the involvement of *cueR* and *pqsH* in stringent regulation of surfing motility in strain LESB58, the Δ*relA*Δ*spoT* mutant was complemented with each cloned gene. Intriguingly, overexpression of both *cueR* and *pqsH* separately restored surfing motility in the surfing-deficient Δ*relA*Δ*spoT* mutant (Fig 3).

## Discussion

Microorganisms move on moist surfaces to enable them to survey new nearby ecological habitats and favourable environmental conditions or to avoid deleterious situations [28]. In

**Table 2. Relative fold-changes of LESB58 Δ*relA*/Δ*spoT* mRNA expression compared to wild-type levels of expression.** 11 regulators were investigated for which PA14 transposon mutant variants exhibited surfing deficiency (no motility or a different form of surfing). The stringent response double mutant was compared to the edges of wild-type motility zones. All experiments were performed on KB agar plates.

| PAO1 Locus Tag | Gene | Product Description | qRT-PCR |
|---|---|---|---|
| PA0034 | | Two-component response regulator | -2.1 |
| PA0520 | *nirQ* | Denitrification regulatory protein nirQ | 7.6 |
| PA1003 | *pqsR* | Transcriptional regulator MvfR / PqsR | 2.0 |
| PA1097 | *fleQ* | Transcriptional regulator FleQ | 2.2 |
| PA1099 | *fleR* | Transcriptional regulator/response regulator FleR | 1.3 |
| PA3477 | *rhlR* | Transcriptional regulator RhlR | 2.3 |
| PA3599 | | Transcriptional regulator | 1.8 |
| PA3921 | | Transcriptional regulator | -2.1 |
| PA4398 | | Two-component sensor | -2.0 |
| PA4726 | *cbrB* | Two-component response regulator CbrB | 3.2 |
| PA4778 | *cueR* | Cu(I)-responsive transcriptional regulator CueR | -4.4 |

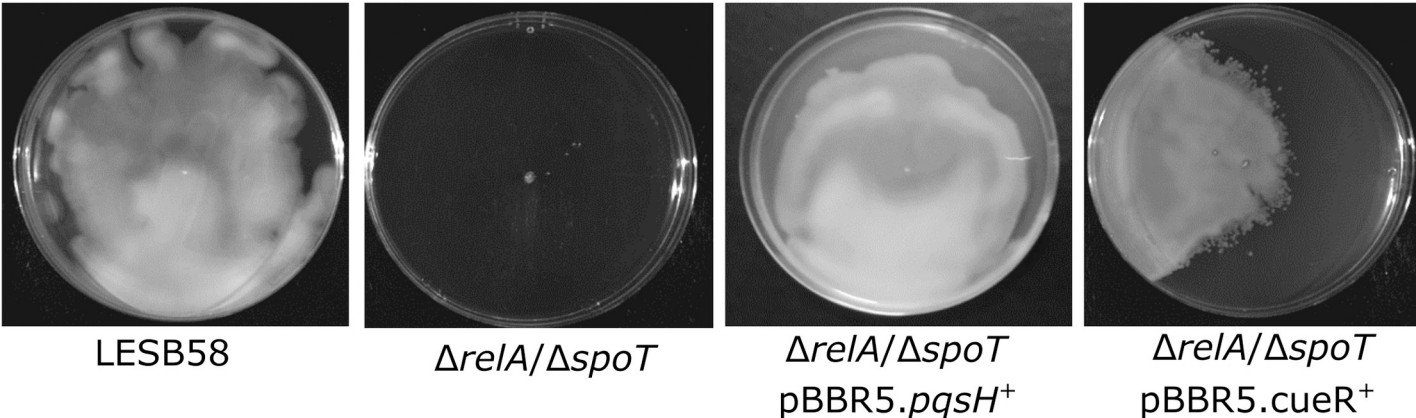

**Fig 3. Surfing motility in *P. aeruginosa* LESB58, Δ*relA*/Δ*spoT*, and effects of overexpression of *pqsH* or *cueR* in the Δ*relA*/Δ*spoT* mutant.** Surfing on KB 0.3% agar plates with 0.4% mucin and the strains were grown for 24 h at 37°C. Experiments were repeated 3–5 times with similar results.

response to specific chemical and physical cues sensed in the environment, microbes can adapt through physiological and morphological differentiation. The stringent stress response is a powerful set of mechanisms that enable bacteria to rapidly adjust to unfavourable environmental conditions and stressful circumstances. Here we demonstrated that the loss of the stringent response in the epidemic *P. aeruginosa* CF isolate LESB58 inhibited a mucin-induced rapid form of surface motility known as surfing. We investigated in more detail the role of ppGpp in surfing motility; a recently identified novel form of motility that requires the presence of mucin that abounds in the lungs of CF patients, contributes to broad-spectrum antibiotic resistance, and has been shown to be conserved in several bacterial species [11, 22].

To investigate the mechanisms underlying surfing impairment in the Δ*relA*Δ*spoT* stringent response mutant, the transcriptional profile of this non-surfing mutant was compared to that of wild-type cells isolated from the edge of a surfing colony. Intriguingly, flagella are required for surfing [11] and most flagellar biosynthesis and chemotaxis genes were upregulated in the non-surfing Δ*relA*Δ*spoT* mutant. This upregulation could be partially due to protein degradation and/or lack of surface flagella. To further identify a mechanism that might explain surfing deficiency, we screened the comprehensive ordered *P. aeruginosa* strain PA14 transposon mutant library, which revealed 224 surfing-deficient mutants. Interestingly, of the 233 mutants affecting swarming motility that were revealed in an analogous screen of the same library [25] there were only 28 mutants that overlapped between the two screens (PA066, *rhlE*, PA0475, *epd*, *aphB*, *gacS*, *fliL*, *hcnC*, *gacA*, *pqsH* (S9 Fig), PA2685, *minD*, *rhlR*, *tli5*, PA3628, PA3631, PA3749, PA4137, PA4144, PA4168, PA4398, PA4616, *cbrAB*, *cueR* (S9 Fig), *pstC*, *cbcV*, and *rmd*). This together with the limited overlap in genes with altered expression between swarming [8] and surfing edge [23] cells, indicates that these two surface motility adaptations, swarming and surfing, are very different.

Further investigation of the expression, in the LESB58 Δ*relA*Δ*spoT* stringent response mutant, of regulatory genes required for surfing revealed that the copper-responsive transcriptional regulator *cueR* was strongly downregulated. Copper is an essential trace element inside bacterial cells, acting as a reactive center for many enzymatic reactions that are important for numerous vital biological processes including respiration, iron uptake and transport, and superoxide detoxification; however too much copper is toxic [29]. In *P. aeruginosa*, CueR has been shown to directly bind to and activate five promoters controlling the expression of 11 genes, namely PA3515 to PA3519, PA3520, efflux operon *mexPQ-opmE*, non-coding RNA

PA3574.1, and a copper homeostasis P-type ATPase *cueA* (PA3920), the last of which was shown to be a virulence factor in a murine model and involved in dissemination to the liver [30]. Of these, PA3518, PA3519 and PA3520 were all upregulated during surfing [23]. Critically we showed here that when *cueR* was overexpressed in a Δ*relA*Δ*spoT* ppGpp-deficient strain, it restored surfing motility. The copper-responsive regulator *cueR* is therefore a new downstream effector of the stringent stress response required for surfing, as well as swarming motility [25].

Transcriptomic analysis also revealed a dysregulation of cell communication, which prompted us to look further into quorum-sensing (QS) signalling pathways. Our previous studies revealed the dependence of surfing motility on all three major QS pathways in *P. aeruginosa* [11, 22]. This was also confirmed by our mutant screen since we identified suppressed surfing motility in mutants in *lasI*, *rhlR*, *rhlI*, *pqsA-E*, *pqsR*, *pqsH*, *phzA1*, and *hcnC*. Interestingly, the expression of the transcriptional regulator *pqsR* and the acyl-homoserine-lactone responsive regulator *rhlR* was 2-fold upregulated in the Δ*relA*Δ*spoT* mutant, but other QS-depended genes, including the *pqsABCDE* operon that includes the quinolone signal response protein gene *pqsE*, *pqsH*, *phnAB*, as well as QS-regulated virulence factors *lasAB*, *aprI*, *aprA/DEFX*, *phzA1/A2/H/M/S*, *hcnABC*, and *ambA/CDE* were all downregulated in the stringent response mutant, by around 2.5-22-fold.

The *Pseudomonas* quinolone signal (PQS, 2-heptyl-3-hydroxy-4-quinolone) pathway depends on PqsR [31], which regulates the expression of the *pqsA-E* operon once activated by 2-heptyl-4-quinolone (HHQ). Further expression of *pqsE* upregulates virulence factors (e.g., *phz*, *hcn*) [31], which is in accordance with our study where we observed a downregulation of all those genes. PQS and *pqsE* have both been suggested to be required for Lectin A production in *P. aeruginosa* [32], however under surfing conditions lectin production appears to depend exclusively on the *rhl* QS system [12] as we found a 7-fold upregulation of *lecA* in the Δ*relA*Δ*spoT*. Together with the upregulation of the *rhl* system we found that two out of three AHL-acylases, *hacB* and *pvdQ*, both hydrolyse long-chain AHLs to prevent excessive signal build-up [33], were downregulated in the stringent response mutant by 1.7- and 3-fold, respectively. Schafhauser *et al*. [34] and Nguyen *et al*. [35] reported that the stringent response negatively regulates 4-hydroxy-2-alkylquinolines (HAQs) by reducing the expression of the *rhl* system, which leads to increased expression of the HAQ biosynthetic gene *pqsA*, the monooxygenase *pqsH*, and the positive regulator *pqsR*. Here, we observed an opposite regulation under surfing conditions, where the *rhl* system was upregulated and HAQ-associated genes were downregulated.

We further focused more closely on the effector functions (signalling molecules) and investigated the quinolone synthase *pqsH* gene that encodes the enzyme involved in the last step of synthesis to produce PQS from HHQ. Overexpression of this gene complemented the surfing deficiency of the Δ*relA*Δ*spoT* stringent stress response mutant. Our previous studies [22] demonstrated that a mutation in the *pqs* operon was surfing deficient and this could be complemented by adding exogenous PQS signalling molecule. This then indicates that the loss of signalling through PQS is one reason why the ppGpp-deficient double mutant Δ*relA*Δ*spoT* is surfing deficient. Thus, our data supports the conclusion that quorum sensing is stringently-controlled to influence surfing motility in LESB58.

Another intriguing feature of the data provided here is the apparent importance of cyclic di-GMP in regulating surfing. We identified four enzymes mediating the regulation of this messenger as being required for surfing motility including the diguanylate cyclases TpbB and SadC and phosphodiesterase FimX and DipA as well as one cyclic-di-GMP binding regulator FleQ (S1 Table). These genes separately regulate other complex adaptive processes including virulence, biofilm formation/dispersion and persistence, a form of adaptive resistance to

antimicrobials, thus, contributing to a relatively large regulon [36–38]. Accordingly, several genes involved in the cyclic di-GMP pathway were found to be dysregulated in the stringent response mutant, including *dipA* which was shown to be essential in mediating surfing motility. Thus, the stringent response and surfing motility in general have a very substantial influence on second messenger activity in *Pseudomonas*.

In summary, we demonstrated that the Δ*relA*Δ*spoT* stringent stress response mutant of the *P. aeruginosa* clinical CF isolate LESB58 had pronounced deficiencies in adapting to environmental stress conditions, suggesting a potential role of the stringent stress response during pathogenesis. We identified two novel downstream effector genes, *cueR* and *pqsH*, both involved in stringent response regulation of surfing motility, and believe it is important to further identify other downstream factors of the stringent stress response, to fully understand its ability to colonize and persist in the CF lung.

## Materials and methods

### Bacterial strains, media, and growth conditions

Bacterial strains used in this study are listed in Table 3 and primers in S2 Table. *Pseudomonas* strains were cultured at 37˚C in LB (Difco), 2xYT (Sigma), King's B medium (KB), or synthetic cystic fibrosis medium (SCFM) [39]. Bacterial growth was monitored at an optical density at 600 nm (OD$_{600}$) using either a spectrophotometer or a 96-well microtiter plate reader (Synergy H1; BioTek). Cultures harboring individual plasmids were maintained by supplementation with 15 μg/ml gentamicin (Gm) for *E. coli* or 500 μg/ml Gm for *P. aeruginosa* LESB58.

### Swarming motility assay

Swarming motility of *P. aeruginosa* strain LESB58 was examined as previously described [40], with slight modifications. Strains were scraped from overnight grown plates and suspended in sterile demineralized water to an OD$_{600}$ of 0.1. Ten μl of a bacterial cell suspension was applied onto KB plates containing 0.35% agar. Plates were then incubated at 37˚C for 48 h. Experiments were repeated at least three times with three replicates for each strain.

### Swimming and surfing motility assay

Swimming motility of *P. aeruginosa* LESB58 strains was examined on KB and SCFM plates containing 0.3% agar, while surfing motility was investigated on the same plates supplemented with 0.4% mucin from porcine stomach (Type II; Sigma-Aldrich). Briefly, *P. aeruginosa* strains

**Table 3. *Pseudomonas aeruginosa* strains used in this study.**

| Strain | Relevant characteristics | Reference |
|---|---|---|
| LESB58 | Liverpool Epidemic Strain isolate | [3] |
| LESB58 Δ*relA*Δ*spoT* | Double deletion mutant of *relA* and *spoT* | [18] |
| LESB58 Δ*relA*Δ*spoT*/*relA*⁺ | Double deletion mutant of *relA* and *spoT* chromosomally complemented with the native *relA* gene including its promoter region; gentamicin$^r$ | [18] |
| LESB58 Δ*relA*Δ*spoT*/ *spoT*⁺ | Double deletion mutant of *relA* and *spoT* chromosomally complemented with the native *spoT* gene including the *rpoZ* and the *rpoZ*-*spoT* promoter region; gentamicin$^r$ | [18] |
| LESB58 Δ*relA*Δ*spoT* (pBBR5.*cueR*⁺) | Double deletion mutant of *relA* and *spoT* transformed with pBBR1MCS-5 containing the *cueR* gene including its promoter region; gentamicin$^r$ | This study |
| LESB58 Δ*relA*Δ*spoT* (pBBR5. *pqsH*⁺) | Double deletion mutant of *relA* and *spoT* transformed with pBBR1MCS-5 containing the *pqsH* gene including its promoter region; gentamicin$^r$ | This study |
| PA14 MAR2xT7 | PA14 transposon mutant library; gentamicin$^r$ | [24] |

were grown to mid-log phase ($OD_{600} = 0.5$), and subsequently spotted (1 μl) onto the respective agar plate, further incubated at 37˚C, and motility zones visually inspected after 15–48 h. The experiments were performed at least three times.

The *P. aeruginosa* PA14 transposon library [24] was screened on SCFM plates containing 0.3% agar and 0.4% mucin. Surfing deficiency was identified either as an inability to spread past the point of inoculation, >70% reduced surface spreading, or a switch to in agar growth/motility which is characteristic of swimming rather than surfing which occurs on the surface.

## Growth curves experiments and pyoverdine production

LESB58 strains were grown overnight (16–18 h) at 37˚C with shaking (250 rpm). Bacteria were pelleted (5000 g, 3 min) and suspended in the respective growth media: KB, dYT, MHB, or SCFM to an $OD_{600}$ of 0.1. Then 200 μl was transferred to a flat bottom 96-well polystyrene microtiter plate (Corning) and incubated at 37˚C with continuous fast linear shaking at 567 cycles per minute (cpm) in a microplate reader (Synergy H1; BioTek). $OD_{600}$ and fluorescence [assessing pyoverdine production [41] at an excitation wavelength of 400 nm and emission wavelength of 460 nm] readings were taken every hour over a 24-hour period. Experiments were performed three times with at least three technical replicates.

## Pyocyanin measurement

Overnight cultures of the LESB58 wild-type, Δ*relA*Δ*spoT* stringent response mutant, and its according *relA* and *spoT* complemented strains, were grown in LB medium, washed in SCFM and resuspended in SCFM at an adjusted $OD_{600}$ of 0.1 and further cultivated at 37˚C with aeration (250 rpm) for 22 h. Pyocyanin was extracted from filter-sterilized supernatants and measured as previously described [42]. Briefly, 2.88 ml chloroform was added to 4.8 ml of culture supernatant and vortexed 10 times for 2 seconds. After centrifugation (14000 rpm, 8 min), 2.4 ml of the chloroform layer was transferred to a fresh tube and 1.2 ml of 0.2 N HCl was added and the mixture vortexed 10 times for 2 seconds. After centrifugation (14000 rpm, 2 min), 1 ml of the top layer (containing pyocyanin) was removed and its absorbance measured at 520 nm ($OD_{520}$). The $OD_{520}$ values were multiplied by the extinction coefficient of 17.072 [42] to obtain the pyocyanin concentrations (μg/ml) of the bacterial supernatants. Experiments were performed three times with at least two technical replicates.

## Adherence experiments

Strains were streaked onto KB agar plates and grown overnight at 37˚C. Bacteria were scraped from the plates and resuspended in KB medium to an $OD_{600}$ of 0.5. One hundred μl of a bacterial suspension was added into polystyrene round bottom 96-well microtiter plates (Falcon) and incubated at room temperature for 1 h. Then each well was washed three times with water, and adhered cells subsequently stained by adding 105 μl of 0.03% crystal violet. Staining was performed on a table top shaker (100 rpm) for 20 minutes at room temperature. Next, plates were washed three times with water and adhered crystal violet dissolved in 110 μl 70% ethanol at room temperature for at least 20 minutes. Absorbance was measured at 595 nm with a BioTek plate reader. Data analysis was performed to calculate the mean and standard deviation, after removal of outliers that were more than one standard deviation from the mean. Data was further normalized to the wild type. Experiments were performed at least three times with up to six technical replicates.

## Biofilm formation under flow conditions and flow-cell imaging

Overnight grown cultures of the wild-type, ΔrelAΔspoT mutant, and complemented strains were cultivated for biofilm formation under flow conditions in chambers as described previously [43]. Briefly, flow-cells were inoculated with *P. aeruginosa* LESB58 strains at an $OD_{600}$ of approximately 0.005 in dYT broth and kept under static conditions for 3 h to enable adherence to the glass chambers after which continuous flow was applied for 3 days at 37˚C. For imaging, the cells were stained with 1 μM SYTO 9, and images captured using a confocal laser scanning microscope (Zeiss, LSM 800) and analyzed using the Zeiss Zen software (v2.3, blue edition). Experiments were performed at least twice.

## Antibiotic susceptibility

The MIC of antibiotics for *P. aeruginosa* LESB58 and stringent response mutants was determined at 37˚C in Mueller-Hinton broth (MHB; Difco) using the broth microdilution assay in 96-well plates [44]. All tests were performed at least three times following the Clinical and Laboratory Standards Institute recommendations. Bacterial growth was examined by visual inspection after 24–48 h of incubation. The MIC was defined as the lowest concentration of a compound that completely prevented visible cell growth.

## Construction of *cueR* and *pqsH* overexpression plasmids

A 501-bp fragment containing the *cueR* gene including its upstream promoter region was PCR amplified using the primers cueR_oe_fwd-Kpn/cueR_oe_rev-Hin. An 1,834-bp fragment containing the *pqsH* gene including its upstream promoter region was PCR amplified using the primers pqsH_oe_fwd-Apa/pqsH_oe_rev-Hin. Each PCR fragment was gel purified and cloned into KpnI/HindIII or ApaI/HindIII restriction sites of plasmid pBBR1MCS-5 [45], yielding pBBR5.cueR and pBBR5.pqsH with additional ability to be expressed from the *lac* promoter. All constructs were sequenced before transformation into *P. aeruginosa* LESB58 ΔrelAΔspoT mutant as previously described [40].

## RNA isolation, quantitative real-time (qRT)-PCR, and RNA-Seq

RNA was isolated from wild-type cells (edge of surfing colony) and the stringent response double mutant (total colony) as previously described [18, 23]. Bacterial cells were collected (six biological replicates) and resuspended in a sterile water mixture with RNAprotect Bacteria Reagent (QIAGEN), isolated using the RNeasy Mini Kit (QIAGEN), and the RNA obtained was DNAse-treated (Ambion/Life Technologies). RNA was quantified using the Synergy H1 microplate reader (BioTek) and RNA integrity determined using the Agilent Bioanalyzer. Three biological replicates (S10 Fig) with the highest RIN score were selected for further analysis. Quantitative Real-Time (qRT-)PCR was performed as described previously [18] at least three times independently using *rpoD* for normalizing transcript levels.

Ribosomal RNA was depleted using the RiboZero Bacteria Kit (Illumina), and cDNA libraries were constructed using the KAPA Stranded Total RNA Kit (KAPA Biosystems). Sequencing was performed by the University of British Columbia Sequencing Consortium using an Illumina HiSeq-2500, generating single end reads (1×100 bp). The read quality of the sequencing samples was checked using FastQC v0.11.6 [46] and MultiQC v1.6 [47]. Alignment of transcriptomic reads to the LESB58 reference genome (obtained from the Pseudomonas Genome Database, ww.pseudomonas.com) was performed using STAR v2.6 [48]. Counts were generated using HTSeq v0.11.2 [49]. Differentially expressed (DE) genes between the double mutant

and wild type were determined using DESeq2 v1.24.0 [50], with thresholds of adjusted $p$-value $\leq$ 0.05 and absolute fold change $\geq$ 1.5 (S1 Data).

### Functional enrichment of DE genes

Enrichment of GO terms were performed using GofuncR, testing the DE genes against a custom set of GO annotations downloaded from the Pseudomonas Genome Database. The full list of 2,584 DE genes was split into up and down regulated, with GO enrichment being performed independently on each of these sets. Results were filtered using a significance threshold of family-wise error rate (FWER) $\leq$ 0.1. Enrichment of KEGG Pathways was done using Gage v2.3.0 on the full list of 2,584 DE genes. Results were filtered for significance based on $q$-value $\leq$ 0.2.

Enrichment of cellular functions, based on manually curated lists was performed on the full list of 2,584 DE genes using Fisher's Exact Test, implemented via a custom script in R v3.6.0 (R Core Team, 2019). Multiple test correction was performed using the Benjamin-Hochberg method and filtered on a significance of $\leq$ 0.05.

### Transmission electron microscopy (TEM)

Bacterial cells were picked from SCFM surfing plates and re-suspended in 10 μl dH$_2$O. Formvar/carbon TEM grids (200 mesh, copper; Ted Pella Inc.) were placed on top of the suspension for 30 s to allow for cell adherence. Excess liquid was removed using filter paper and grids were subsequently stained with 5 μl of 2% aqueous uranyl acetate for 30 s and then washed for 5 s in 10 μl dH$_2$O. Images from multiple grid sections were taken with a Hitachi H-7600 transmission electron microscope (UBC Bioimaging facility).

### Statistical analysis

Statistical evaluations were performed using GraphPad Prism 7.0 (GraphPad Software, La Jolla). *P*-values were calculated using one-way ANOVA, Kruskal-Wallis multiple-comparison test followed by the Dunn procedure. Data was considered significant when $p$-values were below 0.05.

## Supporting information

**S1 Text. The *P. aeruginosa* LESB58 stringent response mutant was impaired in swimming, swarming, adherence, and biofilm formation, as well as being more susceptible to antibiotics.**
(DOCX)

**S1 Table. PA14 transposon mutants that exhibited surfing deficiency.** Surfing deficiency was defined as either no motility, an alternative form of motility, or one-directional motility.
(DOCX)

**S2 Table. Primers used in this study.**
(DOCX)

**S1 Fig. Other phenotypes of *P. aeruginosa* LESB58 wild-type, ΔrelAΔspoT stringent response mutant, and *relA* or *spoT* complemented mutant strains.** A) Swarming on plates with 0.4% agar in KB medium. Plates were incubated for 48 h at 37˚C. Experiments were repeated at least three times with similar results. B) Bacterial growth in KB liquid broth in a 96-well microtiter plate reader at 37˚C under shaking conditions (567 cpm) for 24 h. C) One-hour adherence to plastic in KB medium at room temperature. OD values were normalized to

the wild-type absorbance. * indicates $p$-value $< 0.05$ compared to wild-type. Experiments were performed at least three times. Error bars indicate ± standard error. D) Biofilm formation under flow-cell conditions in dYT broth. Cells were stained after three days for one hour with 1 μM SYTO-9 and subsequently imaged using a Zeiss LSM800 confocal microscope. E) Pyoverdine production (fluorescence: excitation 400 nm; emission 460 nm) after 20 h incubation in KB broth in a 96-well plate under shaking conditions (567 cpm). F) Pyocyanin production after 22 h incubation in SCFM broth under shaking conditions (220 rpm). E, F) Analysis was performed using One-way ANOVA with Dunn correction. **, indicate $p$-value $< 0.01$ compared to wild-type. D-F) Experiments were performed 2–3 times. Error bars indicate ± standard deviation.
(PNG)

**S2 Fig. Swimming phenotypes of *P. aeruginosa* LESB58 and PAO1 wild-type, Δ*relA*Δ*spoT* stringent response mutant, and *relA* or *spoT* complemented mutant strains.** Stringent response mutants and complements were tested under swimming conditions in KB 0.3% agar plates for 24 h (LESB58, top) and SCFM 0.3% agar plates for 15 h (PAO1, bottom) at 37˚C.
(PNG)

**S3 Fig. Transmission electron microscopy images of surfing *P. aeruginosa* LESB58 wild-type and Δ*relA*/Δ*spoT*.** Representative images taken from the centre (left) and edge (middle) of a surfing (SCFM supplemented with 0.4% mucin agar plate) colony, and stringent response mutant (right). Experiments were repeated three times with similar results.
(PNG)

**S4 Fig. Growth curves and adherence of *P. aeruginosa* LESB58 wild-type, Δ*relA*Δ*spoT* stringent response mutant, and *relA* or *spoT* complemented mutant strains in dYT and SCFM broth.** A) Bacterial growth in SCFM liquid broth in a 96-well microtiter plate reader at 37˚C under shaking conditions (567 cpm) for 24 h. B) One-hour adherence to plastic in dYT and SCFM medium at room temperature. OD values were normalized to the wild-type absorbance. * indicates $p$-value $< 0.05$ compared to wild-type. Experiments were performed at least three times. Error bars indicate ± standard error.
(PNG)

**S5 Fig. KEGG pathway—Quorum Sensing *Pseudomonas*.** Visualization of DE genes of the stringent response mutant vs. wild-type under surfing conditions. The quorum sensing pathway (pae02024) was visualized using Pathview. Green boxes indicate a downregulation; red boxes upregulation.
(PNG)

**S6 Fig. Visualization of differentially expressed flagellar genes.** Visualization of DE genes of the stringent response mutant vs. wild-type under surfing conditions. Red bar plots indicate upregulation and green bars downregulation. The asterisks indicate genes with low confidence (adjusted $p$-value $> 0.05$). Dashed line shows significance threshold based on fold change. Gene list was downloaded from KEGG.
(PNG)

**S7 Fig. Visualization of differentially expressed pilus genes.** Visualization of DE genes of the stringent response mutant vs. wild-type under surfing conditions. Red bar plots indicate upregulation and green bars downregulation. The asterisks indicate genes with low confidence (adjusted $p$-value $> 0.05$). Dashed line shows significance threshold based on fold change. Gene list was downloaded from KEGG.
(PNG)

**S8 Fig. Visualization of differentially expressed c-di-GMP genes.** Visualization of DE genes of the stringent response mutant vs. wild-type under surfing conditions. Red bar plots indicate upregulation and green bars downregulation. The asterisks indicate genes with low confidence (adjusted $p$-value $> 0.05$). Dashed line shows significance threshold based on fold change. Gene list was downloaded from https://www.ncbi.nlm.nih.gov/Complete_Genomes/c-di-GMP.html.
(PNG)

**S9 Fig. Surfing motility of *P. aeruginosa* PA14 wild-type, the transposon insertion mutants *cueR* and *pqsH* and their corresponding complemented strains.** Surfing on 0.3% SCFM agar supplemented with 0.4% mucin. All strains were grown for 16–18 h at 37˚C and experiments were repeated at least 3 times with similar results.
(PNG)

**S10 Fig. PCA plot of LESB58 wild-type and stringent response mutant surfing samples.**
(PNG)

**S1 Data. Differentially expressed genes between the double mutant and wild type with thresholds of adjusted $p$-value $\leq 0.05$ and absolute fold change $\geq 1.5$.**
(CSV)

## Acknowledgments

We thank Reza Falsafi for running the bioanalyzer and cDNA library preparation for RNA-Seq runs. We also thank the TEM technician Ross Bradford from the UBC Bioimaging Facility for his help with the imaging.

## Author Contributions

**Conceptualization:** Daniel Pletzer, Evelyn Sun, Michael J. Trimble, Heidi Wolfmeier, Robert E. W. Hancock.

**Data curation:** Daniel Pletzer.

**Formal analysis:** Daniel Pletzer.

**Funding acquisition:** Daniel Pletzer, Heidi Wolfmeier, Robert E. W. Hancock.

**Investigation:** Daniel Pletzer, Evelyn Sun, Caleb Ritchie, Lauren Wilkinson, Leo T. Liu, Michael J. Trimble, Heidi Wolfmeier.

**Methodology:** Daniel Pletzer, Evelyn Sun, Caleb Ritchie, Michael J. Trimble, Heidi Wolfmeier.

**Project administration:** Daniel Pletzer.

**Resources:** Robert E. W. Hancock.

**Software:** Travis M. Blimkie.

**Supervision:** Daniel Pletzer, Heidi Wolfmeier.

**Validation:** Daniel Pletzer, Travis M. Blimkie.

**Visualization:** Daniel Pletzer, Travis M. Blimkie.

**Writing – original draft:** Daniel Pletzer, Evelyn Sun, Robert E. W. Hancock.

**Writing – review & editing:** Daniel Pletzer, Robert E. W. Hancock.

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
