## [Decision Letter · Decision Letter 0]

16 Dec 2019

Dear Prof. Hancock,

Thank you very much for submitting your manuscript "Surfing motility is a complex adaptation that is different from swarming motility and requires the stringent stress response in Pseudomonas aeruginosa LESB58" (PPATHOGENS-D-19-01968) for review by PLOS Pathogens. Your manuscript was fully evaluated at the editorial level and by independent peer reviewers. The reviewers appreciated the attention to an important problem, but raised some substantial concerns about the manuscript as it currently stands. These issues must be addressed before we would be willing to consider a revised version of your study. We cannot, of course, promise publication at that time.

We therefore ask you to modify the manuscript according to the review recommendations before we can consider your manuscript for acceptance. Your revisions should address the specific points made by each reviewer.

(1) A letter containing a detailed list of your responses to the review comments and a description of the changes you have made in the manuscript. Please note while forming your response, if your article is accepted, you may have the opportunity to make the peer review history publicly available. The record will include editor decision letters (with reviews) and your responses to reviewer comments. If eligible, we will contact you to opt in or out.

(2) Two versions of the manuscript: one with either highlights or tracked changes denoting where the text has been changed; the other a clean version (uploaded as the manuscript file).

Additionally, to enhance the reproducibility of your results, PLOS recommends that you deposit your laboratory protocols in protocols.io, where a protocol can be assigned its own identifier (DOI) such that it can be cited independently in the future. For instructions see http://journals.plos.org/plospathogens/s/submission-guidelines#loc-materials-and-methods

We hope to receive your revised manuscript within 60 days. If you anticipate any delay in its return, we ask that you let us know the expected resubmission date by replying to this email. Revised manuscripts received beyond 60 days may require evaluation and peer review similar to that applied to newly submitted manuscripts.

[LINK]

Sincerely,

Matthew C Wolfgang

Associate Editor

PLOS Pathogens

Alan Hauser

Section Editor

PLOS Pathogens

Kasturi Haldar

Editor-in-Chief

PLOS Pathogens

orcid.org/0000-0001-5065-158X

Grant McFadden

Editor-in-Chief

PLOS Pathogens

orcid.org/0000-0002-2556-3526

Reviewer's Responses to Questions

**Part I - Summary**

Reviewer #1: The paper by Pletzer et al entitled " Surfing motility is a complex adaptation that is different from swarming motility and requires the stringent stress response in PA LESB58"

This is a well written paper that demonstrates that mutating the genes required for the stringent response in PA has multiple phenotypes for LESB58 related to surface behaviors. The authors then further explore surfing motility with transcriptional profiling of wt and stringent response mutants as well as transposon mutagenesis to identify genes required for surfing and that these have little overlap with swarming genes. In both of these experiments a large data set is generated and the authors complement cueR and pqsR specifically to show these are down-stream regulators of the stringent response that are required for surfing.

Strengths: Large data set generated with sound experimental design. The role of the stringent response in surfing is clear and further evidence is presented to differentiate surfing and swarming. The significance is that it offers new insight into the role of the stringent response in surface behaviors.

Opportunities: This is primarily a descriptive study with large amounts of genetic data and there is an opportunity as described below to better understand mechanism as well as how the different pieces of data connect. Some of the data such as the antibiotic susceptibility testing doesn't seem to fit with the rest of the manuscript and as an example no mechanism with regard tothe anitbiotic susceptibility is explained from additional transcriptional data or even discussed. Additional control experiments will help understand how much of the transcriptional change is specific to the stringent response defect compared with actual surfing.

Reviewer #2: The study reports on numerous aspects of a clinical isolate of Pseudomonas aeruginosa, strain LESB58, a previously identified CF lung isolate. Primarily, the manuscript reports upon phenotypes for a double- mutant of relA and spoT, where the stringent response is eliminated, are compared to WT and complemented strains. This research is very useful to elucidate how stringent response controls P. aeruginosa behavior.

The short title is a more fitting description than the full manuscript title.

However, the current version of the manuscript requires some work to establish a singular theme.

Reviewer #3: The manuscript entitled: Surfing motility is a complex adaptation that is different from swarming motility and requires the stringent stress response in P. aeruginosa, is an original and well written manuscript with well-planned experiments, careful analysis and novel data on invasion and virulence important in CF. The authors have used a systematic and well-described approach initially starting with phenotypic studies, transcriptome profiling with the LESB58 so-called hyper virulent strain and then moving on to the prototype PA14 strain having a wide pathogenicity profile, and finally qRT-PCR. In a classic analysis and as expected, the authors have used deletion mutants to support their initial observations that the stringent response may be involved. Also, these mutants were foundd to be susceptible to several antibiotics. Very well done and excellent data, overall.

One of the challenges of this type of study is to delineate clearly the surfing phenotype from swarming, twitching and swimming. The authors have shown this previously in different bacterial species using elegant strategies (J. Bacteriol., 2018). Here, the key challenge is the transcriptomics data.

One important item I may have missed: Were the samples done in at least triplicates in the RNASeq analysis? The authors have to address this.

I think the authors have to discuss the ramifications of the impact of 2 to 7.7 fold changes up or down in gene expression when doing transcriptomics in bacteria. It would be an opportunity to educate some of the readers that fold changes in bacteria remains a challenge. This is on page 9 line 2019 and applies even with qRT-PCR.

**Part II – Major Issues: Key Experiments Required for Acceptance**

Reviewer #1: There are experiments and analysis that would significantly strengthen the manuscript.

1) Most of the phenotypes attributed to the stringent response mutants can be attributed to loss or decrease of flagella function and possibly abnormal pilus function. Do these structures change on the cell surface in number or location? Are there more flagella that dont work as well to explain the upregulation of the fle genes? Is twitching motility intact? Are there phenotypic changes in rhamnolipid or exopolysaccharide production that can explain any of these surface changes, If one looks at the cells under the microscope with increasing viscosity what happens to swimming. Phenotypic data (such as transmission electron microscopy) would begin to suggest a mechanism for the changes and also when the CueR and PqsH are overexpressed, to correct surfing these phenotypes can be reexamined under these conditions.

2) The transcriptome compares surfing wt and surfing stringent response. It is unclear how many of these changes are specific to the surfing condition or general to the stringent response regardless of growth condition. I would consider a control experiment, if not for the transcriptome at least for specific genes of interest with RT-PCR 1) wt and stringent response mutant in liquid growth. A comparison of swarming wt and mutant stringent response conditions would also be interesting since under both conditions the phenotype is completely abolished and can you identify surfing specific genes that altered in the stringent response under surfing compared with swarming

3) The relationship between copper metabolism and other transcriptional changes. Quintana et al 2017 JBC show a major change in metabolism under copper stress. Could this explain some of the observed transcriptional changes. What happens to to the other copper homeostatic pathways and if copper levels are manipulated within non toxic ranges. This is another opportunity for mechanism

Reviewer #2: There are inconsistencies in data from the different assays, which make their interpretation complex. These include: 1) The coloration of LESB58 in Fig 1A is differs from that of Fig 1B. Since the authors highlight differences in both pyoverdine and pyocyanin production for the relA/spot mutant as noteworthy, some validation or explanation is needed to better define the basic surfing assay. Similarly, is the strong pigmentation for the spoT-complement singularly noteworthy in comparison to the relA, pqsH, and cueR complements? 2) The swimming assays shown in Figure 2A do not match swimming assays with which I am familiar. A swim phenotype should appear perfectly symmetrical as an expanding circle in the agar. These swim phenotypes are more circular than the swarm images above them, but have many irregularities. Further, the image shown suggests the growth is not uniform and may actually be on top of the agar. Given this, can the authors distinguish between swimming and surfing?

The authors use of different experimental conditions also makes data interpretation difficult. Most assays were conducted using KB medium. However, pyocyanin assays were performed in SCFM, MIC assays were performed in MHB, and flow cell assays used dYT broth. Most importantly, if the authors seek to explain the relation between stringent response and surfing, mucin should be added to all experiments, as this is required for the surfing phenotype.

The current manuscript oscillates between using the surfing phenotype as justification for stringent response experiments and using stringent response as justification for surfing experiments. Individually, many of these results are new and will be of interest to the field. However, the text presents a patchwork of data in search of a theme. Thus, the story requires more definition. If the point is to validate cueR and pqsH as important regulators of surfing, why not show how cueR and pqsH do (or do not) influence swarming, swimming, and antibiotic resistance?

Reviewer #3: One important item I may have missed: Were the samples done in at least triplicates in the RNASeq analysis? The authors have to address this.

I think the authors have to discuss the ramifications of the impact of 2 to 7.7 fold changes up or down in gene expression when doing transcriptomics in bacteria. It would be an opportunity to educate some of the readers that fold changes in bacteria remains a challenge. This is on page 9 line 2019 and applies even with qRT-PCR.

**Part III – Minor Issues: Editorial and Data Presentation Modifications**

Reviewer #1: page 7 table 1 tobramycin resistance is listed for the complemented mutants which are on a gentamicinR plasmid. These are both aminoglycosides and the plasmid level could contribute to tobramycin resistance.

page 9: minor point: How was a defect in surfing defined (what percent of wt diameter, change in morphology etc)

In the discussion new data are introduced and discussed for the first time such as cyclic di-GMP regulon genes. I would move the data to the results and then discuss in discussion

Figure 3. Would consider presenting the metabolic genes as a relational map to understand how the different changes and metabolic pathways relate to each other

Reviewer #2: 1. Line 84, How is the CF lung more stressful than a healthy lung to P. aeruginosa where P. aeruginosa infections are less prevalent? Is increased mucin, itself, an inducer of stringent response? Or is it the use of antibiotics, as a means to treat infections, that stimulates stringent response?

2. Line 212, It would be useful to show the surfing phenotype of PA14 in the supplemental materials.

Reviewer #3: Overall, all the critical experiments have been done. Careful analysis of the data and supplementary figures support this exciting manuscript.

Overall, I see no major issues.

I am looking forward to seeing this in print.

PLOS authors have the option to publish the peer review history of their article (what does this mean?). If published, this will include your full peer review and any attached files.

Reviewer #1: No

Reviewer #2: No

Reviewer #3: No

---

## [Decision Letter · Decision Letter 1]

17 Feb 2020

Dear Prof. Hancock,

Thank you very much for submitting your manuscript "Surfing motility is a complex adaptation dependent on the stringent stress response in Pseudomonas aeruginosa LESB58" for consideration at PLOS Pathogens. As with all papers reviewed by the journal, your manuscript was reviewed by members of the editorial board and by several independent reviewers. The reviewers appreciated the attention to an important topic. Based on the reviews, we are likely to accept this manuscript for publication, providing that you modify the manuscript according to the review recommendations.

Please address the concerns of Reviewer 1 by clarifying the role of flagella in surfing motility. Also, in the revised manuscript, in text references to some of the supplemental data (e.g. Tables S1 and S2) have been removed. Please provide adequate text and references to this data in the manuscript or remove the supplemental information and renumber the existing tables/figures.

Sincerely,

Matthew C Wolfgang

Associate Editor

PLOS Pathogens

Alan Hauser

Section Editor

PLOS Pathogens

Kasturi Haldar

Editor-in-Chief

PLOS Pathogens

orcid.org/0000-0001-5065-158X

Michael Malim

Editor-in-Chief

PLOS Pathogens

orcid.org/0000-0002-7699-2064

Please address the concerns of Reviewer 1 by clarifying the role of flagella in surfing motility. Also, in the revised manuscript, in text references to some of the supplemental data (e.g. Tables S1 and S2) have been removed. Please provide adequate text and references to this data in the manuscript or remove the supplemental information and renumber the existing tables/figures.

Reviewer Comments (if any, and for reference):

Reviewer's Responses to Questions

**Part I - Summary**

Reviewer #1: The revised manuscript "Surfing motility is a complex adaptation dependent on the stringent stress response

in Pseudomonas aeruginosa LESB58" has addressed the majority of questions raised by the reviewers. It is a well written manuscript that describes the roll of the stringent response in surfing motility and adds to the knowledge of Pseudomonas surface behaviors and motility.

Reviewer #2: This manuscript is much improved. The authors have provided very detailed comments to describe their changes in response to comments by all three initial reviews and this is well done. This revised manuscript tells a story that is much more clear.

**Part II – Major Issues: Key Experiments Required for Acceptance**

Reviewer #1: The major experimental issues have been addressed. There is some confusion regarding the data that requires clarification prior to publication regarding the role of flagella regulation in the observed phenotype

Page 4 line 97 "In contrast to the studies of Vogt et al. [14] that suggested that ppGpp was not required for swimming in P. aeruginosaPAO1, as indeed we confirmed (Figure S2), we found that, in the strain LESB58, the stringent

response mutant was further impaired in swimming after prolonged incubation time, but not

completely defective"

The picture in Figure S2 suggests to me it is completely defective. I tried to find a zone of swimming but could not.

The authors then go on to state which is in direct contrast to both the figure and the previous statement.

"There was no basic defect in flagella production due to any of these mutations, since we observed similar swimming zone sizes after 24 h for WT and all mutants in both the strains LESB58 and PAO1 (although the swimming

colony morphology for strain LESB58 was somewhat unusual)"

In the results page 5 they state "We were unable to detect flagella in the spots formed by the relAspoT double mutant (Figure S4), although the large amount of extracellular material under these circumstances made observations difficult.

This information along with the swimming figure has to make one consider whether flagella are present on the surface under these conditions. An up-regulation of flagella genes could be in response to protein degradation and lack of surface flagella. If there is identical swimming in aqueous solution amongst the mutants then this may indeed represent a novel and intriguing form of motility regulation depending on the environment regulated by the stringent response.

Finally on page 13 of discussion this is a misleading sentence.

"One cannot infer anything about surface expression and gene expression and while we did not observe more flagella on the stringent response mutant cells in TEM, a possible explanation could be that the up-regulation of the fle genes could lead to hyper-flagellated cells that might impair motility"

In fact NO flagella were observed on the stringent response mutant cells. One can make no inference about surface protein and flagella structure expression based on gene expression alone. Although I agree that the extracellular material may make it difficult to see flagella, it is still a very sensitive technique. if the bacteria were hyper-flagellate it should be observable on TEM. There is no data to support this possible explanation and more data to support an opposite conclusion.

In summary the regulation of surface flagella and contribution to the observed phenotype is confusing and requires explanation.

Reviewer #2: No new major issues.

**Part III – Minor Issues: Editorial and Data Presentation Modifications**

Reviewer #1: (No Response)

Reviewer #2: None.

PLOS authors have the option to publish the peer review history of their article (what does this mean?). If published, this will include your full peer review and any attached files.

Reviewer #1: No

Reviewer #2: No
---

## [Editor Report · Decision Letter 2]

29 Feb 2020

Dear Prof. Hancock,

We are pleased to inform you that your manuscript 'Surfing motility is a complex adaptation dependent on the stringent stress response in Pseudomonas aeruginosa LESB58' has been provisionally accepted for publication in PLOS Pathogens.

Best regards,

Matthew C Wolfgang

Associate Editor

PLOS Pathogens

Alan Hauser

Section Editor

PLOS Pathogens

Kasturi Haldar

Editor-in-Chief

PLOS Pathogens

orcid.org/0000-0001-5065-158X

Michael Malim

Editor-in-Chief

PLOS Pathogens

orcid.org/0000-0002-7699-2064
---

## [Editor Report · Acceptance letter]

17 Mar 2020

Dear Prof. Hancock,

We are delighted to inform you that your manuscript, "Surfing motility is a complex adaptation dependent on the stringent stress response in Pseudomonas aeruginosa LESB58," has been formally accepted for publication in PLOS Pathogens.

Best regards,

Kasturi Haldar

Editor-in-Chief

PLOS Pathogens

orcid.org/0000-0001-5065-158X

Michael Malim

Editor-in-Chief

PLOS Pathogens

orcid.org/0000-0002-7699-2064